# The Use of Thioflavin T for the Estimation and Measurement of the Plasma Membrane Electric Potential Difference in Different Yeast Strains

**DOI:** 10.3390/jof9090948

**Published:** 2023-09-20

**Authors:** Antonio Peña, Norma Silvia Sánchez, Francisco Padilla-Garfias, Yazmín Ramiro-Cortés, Minerva Araiza-Villanueva, Martha Calahorra

**Affiliations:** 1Departamento de Genética Molecular, Instituto de Fisiología Celular, Universidad Nacional Autónoma de México, Circuito Exterior s/n, Ciudad Universitaria, México City 04510, Mexico; fpadilla@ifc.unam.mx (F.P.-G.); maraiza@ifc.unam.mx (M.A.-V.); mcalahor@ifc.unam.mx (M.C.); 2Departamento de Neurodesarrollo y Fisiología, Instituto de Fisiología Celular, Universidad Nacional Autónoma de México, Circuito Exterior s/n, Ciudad Universitaria, México City 04510, Mexico; yramiro@ifc.unam.mx

**Keywords:** plasma membrane potential, yeast, thioflavin T (ThT), membrane potential indicators, ion transport

## Abstract

The use of the cationic, dye thioflavin T (ThT), to estimate the electric plasma membrane potential difference (PMP) via the fluorescence changes and to obtain its actual values from the accumulation of the dye, considering important correction factors by its binding to the internal components of the cell, was described previously for baker’s yeast. However, it was considered important to explore whether the method developed could be applied to other yeast strains. Alternative ways to estimate the PMP by using flow cytometry and a multi-well plate reader are also presented here. The methods were tested with other strains of *Saccharomyces cerevisiae* (W303-1A and FY833), as well as with non-conventional yeasts: *Debaryomyces hansenii*, *Candida albicans*, *Meyerozyma guilliermondii*, and *Rhodotorula mucilaginosa*. Results of the estimation of the PMP via the fluorescence changes under different conditions were adequate with all strains. Consistent results were also obtained with several mutants of the main monovalent transporters, validating ThT as a monitor for PMP estimation.

## 1. Introduction

Long ago, the mechanism of potassium (K^+^) transport in *Saccharomyces cerevisiae* was proposed [1]. At the plasma membrane of the yeast, the H^+^-ATPase (Pma1p) acidifies the external medium by pumping out protons (H^+^), thus alkalinizing the cell interior, which consequently produces an electric plasma membrane potential difference (PMP). Trk1p, the main K^+^ transporter in yeasts, and Trk2p, which functions to a lesser extent and under certain physiological conditions, both pump K^+^ inside the cell, taking advantage of the potential previously established by Pma1p and thus maintaining electrical and ionic homeostasis in the cell. In addition, the Tok1p transporter is an outwardly rectifying K^+^ channel that is activated under conditions of plasma membrane depolarization [2] (Figure 1). Previously, it was reported that the Pma1p is 20 to 50% stimulated by the addition of KCl, increasing the H^+^ pump outside the cell and subsequently increasing the PMP [3].

A similar mechanism was also found in *Neurospora crassa* [8] and, later, in other yeasts and fungi [9], as well as in plants [10]. When the changes of the external and internal pH were measured [1], it was shown that, in fact, while the external pH decreased, the internal pH of the cells increased [11]. Those studies were then followed by different attempts to estimate or measure the plasma membrane potential difference (PMP), originally via the accumulation of tetraphenylphosphonium [12,13,14].

Attempts have been made to estimate the PMP by using fluorescent probes, first with ethidium bromide, but this dye was found to be a competitive inhibitor of the K^+^ transporter [15]. The monitor DiSC_3_(3) (dipropylthiacarbocyanine iodide) was then found to be adequate to estimate the changes of the PMP by following its fluorescence changes under different conditions [16]. Other attempts to estimate the PMP have also been performed by measuring either the fluorescence changes or the accumulation of different cationic molecules [13,16,17,18,19,20], but most of those results appear to be inaccurate, mainly because they did not consider that besides the uptake of the molecules via effect of the PMP, a part of them was also accumulated due to their binding to the internal components of the cell. 

More recently, the use of a dye to estimate, by fluorescence, the PMP changes in yeast, as well as the actual values by its accumulation, was reported [20]. The dye used was first supposed to be acridine yellow, previously purified from an old flask, but when attempts at further experiments were conducted with two new acridine yellow dyes, obtained from Sigma and Pfaltz and Bauer (as was the originally used dye), the data could not be reproduced. NMR and mass spectra showed that the dye originally used was really thioflavin T (ThT) (Figure 2), a stain used in histological studies to show amyloid fibrils and in flow cytometric analysis of human reticulocytes [21]. A retraction note made clear that the dye used was not acridine yellow, but thioflavin T (ThT) [22]. 

The structure of ThT (Figure 2) consists of a dimethylated benzothiazole ring coupled to a dimethylamino benzyl ring [23]. In solution, the two rings act as a molecular rotor, and the rings; rotation causes the free ThT to have a low fluorescence which increases when the viscosity of the medium is increased [23,24,25] or when ThT binds to biomolecules such as proteins, DNA, RNA, and lipids [26,27,28,29].

The pure dye was then obtained from Biotium, and similar results were obtained with some modifications to the methods employed [30]; more recently, two independent groups have used ThT to estimate the PMP in yeast [31,32].

Several authors, including our group [13,16,17,18,19,20], measured the accumulation of different probes but did not consider that the dyes with a cationic and hydrophobic nature not only accumulate due to the PMP but also bind to anionic and hydrophobic molecules inside the cells. The improved method then included the estimation of these factors, which is necessary for correcting the non-corrected accumulation values [30]. 

All our former studies were performed with *S. cerevisiae* baker’s yeast (Azteca), but the doubt persists as to whether this method can be applied to other strains from different species. In this work, we present the data obtained with other strains: *S. cerevisiae* strains W303-1A and FY833, as well as *Candida albicans*, *Debaryomyces hansenii*, *Meyerozyma guilliermondii*, and *Rhodotorula mucilaginosa*. The suggested methodology was validated by also using *S. cerevisiae* mutants of the main K^+^ transporters, such as Trk1p, Trk2p, and Tok1p, and via different methodological approaches.

The results confirmed that ThT can be used to obtain actual values and changes of the PMP. Among its properties, this dye is characterized by a lower hydrophobicity than DiSC_3_(3) [16].

## 2. Materials and Methods

### 2.1. Strains and Growth Conditions

The yeast cells used are enlisted in Table 1. Strains *S. cerevisiae* W303-1A and TOW (*tok1Δ*) (kindly donated by Dr. Hana Sychrová) were maintained in 2%-agar plates of YPD medium (1% yeast extract, 2% peptone, 2% glucose) with 80 mg·L^−1^ adenine and 30 mg·L^−1^ uracil; *S. cerevisiae* FY833 in YPD were maintained in medium supplemented with amino acids plus uracil (L-histidine, L-tryptophan, L-methionine, uracil, at a final concentration of 20 mg·L^−1^; L-lysine, 30 mg·L^−1^ and L-leucine at 100 mg·L^−1^); mutants *trk1Δ*, *trk2Δ* and *trk1Δ trk2Δ* were maintained in YPD medium supplemented with 100 mM KCl and amino acids plus uracil; and other strains were maintained in YPD. Cultures were renewed every month.

Cultures were started by inoculating the cells in 500 mL of liquid of their respective medium, as indicated previously, for 24 h at a constant temperature (30 °C, except for *D. hansenii*, which was cultured at 28 °C) in an orbital shaker at 250 rpm. In order to be able to observe the effects of substrates to energize the cells, they were starved by washing them via centrifugation, suspending them in 250 mL of water, and incubating them in the same orbital shaker for 24 h, except in the case of *C. albicans*, *D. hansenii*, *M. guilliermondii*, and *R. mucilaginosa*, which were fasted for 48 h. The cells were then collected via centrifugation, washed once with water, and suspended in water at a ratio of 0.5 g (wet weight, w. w.)·mL^−1^.

### 2.2. Membrane Potential Estimations, Fluorescence of ThT

Experiments were conducted with variations on the procedure reported before [17]. The basic medium contained 10 mM MES-TEA (2-(N-morpholino)ethanesulfonic acid adjusted to pH 6.0 with triethanolamine) in a final volume of 2.0 mL, which is sufficient to allow for uniform mixing in the cells. The medium found to give the best results contained 10 μM BaCl_2_ and 20 mM glucose. Tracings were started by adding 50 mg (w. w.) of cells and then 15 μM ThT (Biotium). When the cells reached anaerobiosis (O_2_ = 0), as indicated by a small fluorescence increase, 5 µL of 3% H_2_O_2_ (as a mild source of O_2_) was added, then 10 µM CCCP (carbonyl cyanide m-chlorophenylhydrazone), then finally, 10 mM KCl. Fluorescence was followed at the corresponding maxima of both excitation and emission wavelengths 470–505 nm. An SLM Aminco spectrofluorometer updated by Olis with stirring and temperature regulation at 30 °C was used. Slits were fixed at 8 nm.

### 2.3. Fluorescence Measured with a Multi-Well Plate Reader

Experiments were performed under similar conditions to the ones used in the fluorometer. The relative fluorescence changes of the different wild-type and mutant yeasts were followed with a FLUOstart Omega 96 multi-well plate reader using excitation at 485 nm and emission at 520 nm, with constant agitation at 30 °C. Experimental conditions were as follows: (1) 10 mM MES-TEA buffer, pH 6.0, 10 µM BaCl_2_, 20 mM glucose, and 15 µM ThT, 200 µL final volume. At 5 min from the beginning of the trace, 5 mg of cells (w. w.) were added, and the readings were continued for 35 min more. From the final one, the basal fluorescence was subtracted (fluorescence 40 min − fluorescence 5 min = Δ); this value (Δ_1_) was taken as 1.0. (2) To the same initial mixture, 10 µM CCCP was added, and in (3), the same initial mixture plus 10 µM CCCP and 20 mM KCl. Data collection was similar as in condition (1); but the final values (Δ_2_, Δ_3_) were compared to Δ_1_ to obtain a fold increase value.

### 2.4. Microscopy

The cells, 25 mg (w. w.), were added to a mixture of 10 mM MES-TEA buffer, pH 6.0, 10 µM BaCl_2_, 20 mM glucose, and 15 µM ThT, at 1.0 mL final volume. In the same mixture, additions of 10 µM CCCP, followed by 10 mM KCl, were included.

An LSM 710- Zeiss microscope with the Software Zen black 2.3 was used. The images were obtained via confocal microscopy using a 63X WPlan-Apochromat objective in water immersion with a 1.0 numeric aperture (NA) and 2.1 mm work distance. The conditions were as follows: laser: 458 nm excitation, 463–550 nm emission, with 30% of the laser intensity; zoom: 2× equivalent to 126× magnification. 

In all strains used, it was found that in all conditions, the dye was not distributed inside the whole cell, as will be described in the Section 3. This, in fact, was taken into consideration to measure the vacuole/cell ratio (used to calculate the internal water content) from the microscopic images by means of Fiji software [39]. For microscopic images, samples of 3-to-4 biological replicates were used for each strain, and between 60 and 300 data from each were used to measure the vacuole/cell volume ratio.

### 2.5. Membrane Potential Measurements by the Accumulation of ThT

The medium used to measure the accumulation of this dye was also 10 mM MES-TEA buffer, pH 6.0, 10 μM BaCl_2_, 20 mM glucose, 100 µM ThT, 150 mg (w. w.) of yeast, in a final volume of 3 mL. The tubes were incubated for 10 min at 30 °C. Different conditions of incubation were as follows: (a) with glucose alone; (b) with glucose plus 10 µM CCCP; (c) glucose plus CCCP and adding 100 µg of chitosan-HCl to permeabilize the cells [40]. After 10 min, the cells were centrifuged, and the supernatant was collected to measure, in adequate dilutions, the concentration of ThT via its absorbance at 411 nm (Beckman DU650 Spectrophotometer). Another set of tubes, one with glucose and CCCP, and another with glucose, CCCP, and chitosan, were incubated for 10 min, and then, to those with glucose and CCCP, 10 mM KCl was added, and 200 mM KCl was added to those with chitosan. After 10 min more, the cells were centrifuged, and the supernatant was collected. The concentration of ThT was then also measured by its absorbance at 411 nm. The absorbance values were then compared with those of a concentration curve from 10 to 40 µM ThT to calculate the concentration of the dye remaining in the supernatants. This latter value was multiplied by the external volume (2.85 mL) to obtain the total dye amount remaining outside the cells. This value was then subtracted from the total amount added (300 nmoles) to obtain that inside the cells. Next, the internal amount of the dye was divided by the internal water content of the cells, excluding the vacuole, which in all cases excluded the dye, most probably because of its positive potential difference, obtained for each strain by the inulin test according to [30,41], and considering the apparent vacuole/cytoplasm ratio observed via microscopy and described in Section 2.4. This gave the value of the internal concentration of the dye, which, divided by the external one, and applying the Nernst equation, results in the apparent value of the plasma membrane potential:$$∆\psi =-2.3026\log\;\frac{RT}{zF}*\;\left(concentration\;ratio\right)=-\log\frac{\left[int\right]}{\left[ext\right]\;}\;*\;60.12$$
where:Δ*ψ* is the plasma membrane potential difference, measured in volts (V); *R* is the universal gas constant, (8.314 J·K^−1^·mol^−1^); *T* is the temperature in Kelvin degrees (30 °C = 303 K); *z* is the number of elementary charges of thioflavin T (ThT) (=1); *F* is the Faraday constant, equal to 96,485 J·V^−1^·mol^−1^;[*int*] is the intracellular concentration of ThT; [*ext*] is the extracellular concentration of ThT; 2.3026 RT/F = 60.12 mV (considering our experiments at 30 °C).

These values in the cells incubated only with glucose do not consider the dye accumulated by the mitochondria, nor that accumulated by its simple adsorption via the internal components of the cell, both by its cationic and hydrophobic nature. 

The dye accumulated by the mitochondria was eliminated by the addition of 10 µM CCCP (a concentration that uncouples mitochondria, but K^+^ transport is not affected) [17]. Then, the total amount of dye remaining inside the cells after the addition of chitosan represents that which is bound due both to its hydrophobic and cationic nature, not to the membrane potential, and that remaining after the addition of chitosan and 200 mM KCl (a higher concentration of KCl displaces the dye attached to anionic molecules) gives the amount which remains inside only due to its hydrophobic character. The difference between the amount remaining inside only after the addition of chitosan, minus that remaining after the addition of both chitosan and KCl, gives the amount of ThT taken by the cells independently of the PMP. The values obtained were subtracted and considered in the calculations as the “corrected value”.

### 2.6. Flow Cytometry

Experiments were performed under similar conditions as those used for the microscopic observations (Section 2.4), but including, after the addition of 10 mM KCl, the addition of 50 µg chitosan–HCl and then 200 mM KCl. The fluorescence changes of the different wild-type strains and mutants were followed by flow cytometry using the Attune^®^ Acoustic Focusing Cytometer (Applied Biosystems). Excitation was attained with a violet laser of 405 nm and emission with a green filter of 507–537 nm (channel VL2-A).

The relative fluorescence mean values in channel VL2-A were calculated with Attune^®^ Cytometric Software v1.2.5. It is important to mention that since each strain displayed different values, a value of 1.0 was defined for the fluorescence emitted by the sample with only ThT and yeast. Subsequently, the relative fluorescence of the samples with CCCP (10 µM) and CCCP + KCl (10 mM) was reported as a fold increase value regarding those with only ThT. In all cases, representative results from three or more similar experiments are shown.

### 2.7. K^+^ Uptake

This parameter was followed by using a specific electrode connected to a pH meter and an acquisition system. The medium used was 2 mM MES-TEA buffer, pH 6.0, 20 mM glucose, 10 μM KCl. Tracings were started by the addition of 100 mg of the cells (w. w.) in a final volume of 10 mL. When indicated, variable concentrations of CCCP were added. A standard curve for the K^+^ tracings was performed by the successive addition of 0-to-50 μM KCl.

### 2.8. Oxygen Consumption

Oxygen consumption by the cells was measured with a Clark electrode connected to a polarization and measurement device (YSI, Yellow Springs Instrument, Co., Yellow Springs, OH, USA), collecting the data with a voltmeter linked to a computer. The medium, 5.0 mL final volume, contained 10 mM MES-TEA buffer, 20 mM glucose, and 50 mg cells (w. w.). Where indicated, different concentrations of CCCP were added. Experiments were conducted in a closed chamber under continuous stirring and maintained at 30 °C.

### 2.9. Statistical Analysis

All values are represented as averages ± standard deviation (SD) of independent biological cultures (3–13, depending on the experiment). Statistical analysis was performed with GraphPad Prism (version 8.0.1, San Diego, CA, USA), one-way and two-way analyses of variance (ANOVA) with different post hoc analyses (Dunnett’s, Tukey’s, and Sidak’s, as indicated in each figure and table caption), as well as a Student-*t* test, were conducted, all of them with a *p*-value of 0.05.

## 3. Results

### 3.1. Estimation of the PMP by Changes in the Fluorescence of ThT with Different Yeast Strains 

ThT enters the cell and fluoresces because of its accumulation inside. As shown in Figure 3, with *S. cerevisiae* Azteca, *C. albicans*, *D. hansenii*, *M. guilliermondii*, and *R. mucilaginosa*, upon the addition of cells, an increase in the tracing was observed, and another one after adding ThT, which was stabilized after around one minute, and followed by a small increase when the cells reached anaerobiosis (O_2_ = 0, at different times for each strain, as can also be seen in Appendix A). This was verified by observing a decrease in the previous level by adding 5 µL of 3% H_2_O_2_ as a source of O_2_. Then, the addition of 10 µM CCCP resulted in a large increase in the signal, and the further addition of 10 mM KCl produced a variable decrease in fluorescence in accordance with a depolarization of the plasma membrane due to K^+^. *D. hansenii* took around 10 min to achieve anaerobiosis.

It was important to define whether these changes could be validated with the mutants of the main K^+^ transporters in yeast, so we made the same experiment in *trk1Δ*, *trk2Δ* the double *trk1Δ trk2Δ* mutants, as well as in the TOW (*tok1Δ*) mutant, with their respective wild-type genetic backgrounds as controls (*S. cerevisiae* FY833 and *S. cerevisiae* W303-1A). As shown in Figure 3, there was a diminution in the response of the dye to the addition of KCl in the *trk1Δ* and a lack of response to KCl in the double-mutant *trk1Δ trk2Δ* as would be expected. With the TOW mutant, lacking the potassium exit channel, we could observe a gradual diminution of the fluorescence of ThT, indicating a progressive depolarization of the strain. In this mutant, the response to KCl was larger than in its wild type, indicating a larger depolarization due to the cation. 

### 3.2. Results Obtained by Fluorescence in a Multi-Well Plate Reader

The same experiment performed in the fluorometer was tested measuring the changes of fluorescence in a multi-well plate reader using a FLUOstart Omega apparatus at 480–520 nm, as explained in the Section 2.

Experiments were performed in the same incubation buffer as before. At 5 min after the beginning of the tracing, 5 mg of cells (w. w.) were added, and the fluorescence in each well was recorded for 35 more min (Figure 4a,b). From the final one, the basal fluorescence was subtracted (fluorescence 40 min − fluorescence 5 min = Δ), and this value (Δ_1_) was taken as 1.0. To the same initial mixture, 10 µM CCCP was added, and in another set of experiments, the similar initial mixture plus 10 µM CCCP and 20 mM KCl were added. Data collection was alike at the condition with only glucose plus ThT and yeasts, but the final data (Δ_2,_ Δ_3_) were compared to Δ_1_ to obtain a fold increased value (Figure 4c–f).

For the double-mutant *trk1Δ trk2Δ*, as expected, no change with KCl was observed since without the main K^+^ transporters, no potential could be dissipated. 

In the TOW mutant, no statistically significant changes could be seen as compared to its wild type; however, a tiny trend to depolarize can be observed if compared to its wild-type W303-1A, better seen in the individual data. 

### 3.3. Fluorescence Images of the Cells

In order to test our assumptions from the fluorescence changes, microscopic images were obtained. As a control, Figure 5 shows that in the cells from *S. cerevisiae* (Azteca strain), in the presence of glucose alone, the dye mostly accumulated in the mitochondria, showing a low fluorescence in the cytoplasm. Then, the addition of CCCP in a concentration where only mitochondria are uncoupled (10 µM) produced a large increase in fluorescence because the dye is now distributed in the cytoplasm, in a larger volume. The further addition of 10 mM KCl produced a significant decrease, most probably because this produces an efflux of the dye due to the partial decrease in the PMP.

Similar results were obtained with the other strains used, with some differences: a low fluorescence with glucose alone then an increment upon the addition of CCCP, as well as the further decrease after the addition of KCl (Figure 6). Higher zoom images for all strains of the glucose condition are displayed in Appendix A.

### 3.4. Values of PMP Obtained from the Accumulation of ThT

As already mentioned, previous work by other authors and ourselves [11,12,13] tried to measure the PMP by following the accumulation of cationic molecules; it was shown that ThT can act as a Nernstian membrane potential indicator, as had also been shown in bacteria [42], but first of all, they (and we) did not consider what is shown in Figure 5 and Figure 6—that is, the accumulation of the cationic agents by the mitochondria. However, the following figure shows an additional factor involved in the accumulation of the dyes used, which, in this case, is the unspecific binding of ThT (or any hydrophobic, cationic dye) to the internal molecules of the cell (Figure 7). A few years ago, we described a method with which to actually measure the PMP, based on the accumulation of ThT, that considered this circumstance [20]. Several aspects had to be taken into consideration and solved, and they are as follows.

1. The dye entering the cells not only accumulates in the cytoplasm but also, due to the membrane potential of the mitochondria, enters these organelles, where the high concentration attained results in the quenching of the fluorescence (Figure 7a).

This problem was eliminated by using a low concentration of an uncoupler (CCCP, 10 µM) to collapse the mitochondrial membrane due to the H^+^ gradient that did not affect the plasma membrane gradient (Figure 7b). 

While on the subject, these fluorescence signals may be used to obtain changes in mitochondrial membrane potential and estimate the changes from an initially defined membrane potential [43].

However, it is important to point out that in order to eliminate the accumulation of the dye by the mitochondria, it is necessary that the cells have in the plasma membrane a lower sensitivity to the uncoupler to that in the mitochondria, which appears not to be the case for all the strains tested, as it will be shown further below.

2. Because the dye, in the presence of a low concentration of CCCP, accumulated in the cytoplasm and not in the vacuole, in order to obtain its concentration, the value of the cytoplasmic volume excluding the vacuole had to be obtained. This was achieved by measuring the cell volume by the distribution coefficient of ^14^C-inulin and via the following: (a) subtracting the dry weight of the cells to obtain the total internal cell water; and (b) by means of the microscopy images, obtaining the vacuole/cell volume ratio to calculate the actual value of the cytoplasmic volume and that excluding the vacuole, as described in the Section 2. The values obtained for the cytoplasmic water volume of the 150 mg of cells of the different strains are presented in Table 2.

3. However, as shown in the scheme of Figure 7, because of its hydrophobic and cationic nature, a significant part of the dye used is taken up by the cells due to its binding to the molecules inside the cell. This problem was solved by measuring the remaining dye inside the cells after their permeabilization with chitosan (Figure 7c), which gave the total amount bound, or after the permeabilization plus the addition of a significant concentration (200 mM) of KCl, which gave the amount remaining inside the cells only because of its hydrophobic nature (Figure 7d). Subtracting the values of the hydrophobic concentration from the total values obtained, the net concentration of the dye under different conditions was obtained, which, divided by the external concentration obtained directly, allowed for the actual and most approximate values of the PMP using the Nernst equation.

The values of Table 3 and Table 4 show that, in fact, reasonable values for the PMP could be obtained by measuring the accumulation of ThT. These values were, of course, lower when CCCP was added, which produces the efflux of the dye accumulated by the mitochondria, and even more so after the addition of 10 mM KCl, by which K^+^, due to its positive charge, by entering the cell, produces a decrease in the PMP. However, particularly with *D. hansenii*, *M. guilliermondii*, and *R. mucilaginosa*, the values obtained by correcting the data after permeabilization with chitosan produced strange, corrected values, i.e., higher with K^+^. This can be explained by the higher sensitivity of the plasma membrane that these three strains have to CCCP (Appendix A, explained below in Section 3.5). 

Table 3 shows that when cells are incubated with glucose (basal PMP), most of the different yeasts have similar PMP values, except for *C. albicans* and *S. cerevisiae* Azteca, which have the highest PMP values, while *D. hansenii* and *R. mucilaginosa* have the lowest values. This observation is supported by the statistical analyses performed, which show that the value of *C. albicans* is different for all strains, apart from *S. cerevisiae* Azteca, while *S. cerevisiae* Azteca presented only a significant difference with the values obtained for *D. hansenii* and *R. mucilaginosa*.

Comparing the basal PMP values obtained with glucose (corrected and non-corrected) in Table 4, it can be observed that the low-affinity K^+^ transporter mutant strain (*trk2Δ*) presented a PMP similar to that of strain *S. cerevisiae* FY833, while the high-affinity K^+^ transporter mutant strain (*trk1Δ*) and the double-mutant strain (*trk1Δ trk2Δ*) had a similar PMP, which is very high, as expected, when compared to strain *S. cerevisiae* FY833, an observation that was statistically confirmed. Nevertheless, the TOW strain, lacking the Tok1p transporter, showed no significant difference when comparing its calculated PMP (correct and non-corrected) with that of its *S. cerevisiae* W303-1A genetic background.

The depolarization percentage by the addition of K^+^ is more than 50% in the wild-types (Figure 8), whereas in the *trk1Δ* and *trk1Δ trk2Δ* mutants, as expected, the depolarization percentage of the PMP was low, with *trk1Δ trk2Δ* being the lowest, due to the fact that K^+^ could not enter the cell; this was also observed in the measurement conducted in the ThT accumulation experiments, and also in those using the multi-well plate reader and flow cytometry, as shown later in Figure 9. 

Tok1p controls the K^+^ release from yeast cells and maintains a balance in the PMP, so *tok1Δ* leads to a higher membrane depolarization. The lack in ion channel expression correlates with changes of PMP, as shown by the increased fluorescence, due to the higher PMP reached because, in the absence of this channel, the cells do not have the opportunity to expel the monovalent cation, resulting in an increased fluorescence. This could not be seen compared with its wild-type in this graph, but it was clear in Table 4 with respect to their plain respective PMP values. 

### 3.5. ThT Fluorescence as a Monitor of PMP, Can also Be Measured by Flow Cytometry

With flow cytometry being a powerful tool for detailed analysis of complex populations in a short period of time, it can be an important tool with which to obtain and confirm the results of the PMP estimation via fluorescence, as well as those obtained via microscopy (Figure 9). 

A basal fluorescence with ThT and glucose was observed; then, the addition of CCCP to uncouple mitochondria produced a large increase in fluorescence. The further addition of 10 mM KCl resulted in a significant decrease, most probably because this produces an efflux of the dye. Permeabilization with chitosan produced an increment of fluorescence that was quenched by the addition of a higher concentration of KCl (Figure 9a,b). 

All the wild-type strains had a significant increase in fluorescence with CCCP and a depolarization when K^+^ was subsequently added (represented as a decrease in fluorescence), resembling what was obtained with the previously described techniques. The use of mutants in the main K^+^ transporters allowed us, in these experiments also, to test the reliability of ThT as a monitor of PMP, obtaining what was expected for each case, i.e., increment in the fluorescence level as result of the addition of CCCP and, depending on the mutant, a higher fluorescence for the *trk1Δ* strain and for the double-mutant *trk1Δ trk2Δ*, resembling a lower response to the KCl added. Again, a difference in the response to KCl was not possible to observe in the TOW strain, as compared to its wild-type. 

These results demonstrate that the changes in ThT fluorescence are also measurable and easy to detect via flow cytometry in a reliable way. 

### 3.6. K^+^ Uptake and Oxygen Consumption

It was important to determine a concentration of CCCP that dissipated the mitochondrial potential without affecting the PMP; in this way, all the strains were tested for K^+^ transport and oxygen consumption in the presence of different concentrations of the uncoupler.

When measuring the K^+^ transport with a specific electrode, the addition of the cells to the medium produced the efflux of significant amounts of K^+^ followed by its uptake, almost to the initial levels in some strains such as *S. cerevisiae* Azteca, *S. cerevisiae* FY833, and *trk2Δ* mutant (Appendix A). Then, the addition of CCCP (2.5, 5.0, or 10 µM) in most of the strains did not affect K^+^ transport, except for *D. hansenii, M. guilliermondii*, and *R. mucilaginosa*, indicating a higher relative sensitivity of the plasma membrane potential to even low CCCP concentrations in these three strains. It is clear that in *D. hansenii*, not only 10 µM but even 5 µM CCCP could reverse the K^+^ transport, pointing to a much higher sensitivity of the PMP to the uncoupler. *M. guilliermondii*, on the other hand, could be partially inhibited with 10 µM, but not with 5 µM CCCP. *R. mucilaginosa* is extremely sensitive to CCCP, provoking the inhibition of K^+^ uptake even at 2.5 µM (Appendix A). Although, in other yeast strains, 10 µM CCCP had no effect on K^+^ uptake, this provided the awareness that some of the strains have different sensibility to this ionophore. As expected, the double mutant of Trk1-2p transporters did not show K^+^ uptake.

The inhibition of K^+^ uptake by *D. hansenii* and *M. guilliermondii* with 10 µM CCCP, and also *R. mucilaginosa* with less CCCP, might set in doubt the results of the PMP measurements in which this concentration was used. However, it has to be considered that in the ThT accumulation experiments, 1.5 times more cells were used (150 mg). Because of this, the cells may require a higher concentration of the uncoupler in order to affect the plasma membrane. In fact, an experiment was performed in which the accumulation of the dye was measured using the usual protocol but comparing the results with 10 or 5 µM CCCP. The results are shown in Table 5, where it can be seen that values are almost similar under both conditions, particularly for those with CCCP or CCCP plus KCl.

Oxygen consumption is an important parameter by which to check the electrochemical gradient dissipation by CCCP, even as low as 2.5 µM of which can stimulate respiration in some of the strains. On the other hand, 20 µM CCCP inhibited respiration in *S. cerevisiae* Azteca, W303-1A, FY833, and *D. hansenii* (Appendix A), even though K^+^ transport seems not to be affected with this concentration.

## 4. Discussion

Antecedents of the use of ThT to estimate yeast PMP by its fluorescence changes, as well as obtaining the actual values by measuring its accumulation, and correcting by its binding to the internal components of the cell, were described previously in baker’s yeast cells [30]; however, some doubts remained concerning the application of this methodology in other conventional and non-conventional yeast strains, as well as in mutants of the different transporters involved in monovalent cations’ (mainly K^+^) influx and efflux.

The current methodology proposed in this work to obtain actual values of the PMP in seven wild-type strains and four mutants of the K^+^ transporters, in general terms, implies measuring the following: (a) the accumulation values of the dye (ThT) taken by the cells; (b) the elimination of its accumulation in the mitochondria by the addition of an uncoupler, whose concentration may dissipate the membrane potential of these organelles but not of the plasma membrane (Appendix A); and (c) the actual concentration of ThT in the cytoplasm. All those effects are summarized in Figure 7 and verified with different methodologies, as shown in Figure 3, Figure 4, Figure 5, Figure 6 and Figure 9. 

As we mentioned above, mitochondria could accumulate ThT (as shown in Figure 5, Figure 6 and Appendix A); this effect was also previously demonstrated in mammalian cells treated with ThT to determine the mitochondrial membrane potential [44]. Besides the adequate concentration of CCCP, it is also important to consider that if a low number of cells are used with high concentrations of CCCP, the plasma membrane may be affected, and the PMP data would no longer be adequate. This effect was previously reported by Farkas et al., where they suggested that excessive uncoupling with FCCP rapidly collapses the mitochondrial membrane potential and, subsequently, the PMP in mammalian cells [45].

Moreover, due to its hydrophobic and cationic nature, the dye, besides being taken up due to the PMP, binds to the internal hydrophobic and anionic components of the cell such as proteins, lipids, and RNA [17,20,26,30,46,47]. We could determine the value of dye bound to the internal components of the cells by permeabilizing yeast with a polycation such as chitosan [40], which allows for the detection of the total amount of dye taken up by the cells, independently of the PMP. Then, by adding a high concentration of KCl (200 mM), that which was bound due to its cationic nature was displaced. Finally, the difference between the total internal dye in the cell and the one displaced by a high concentration of KCl provided the actual value of the ThT entering the cell only due to its cationic nature.

First of all, we compared the fluorescence changes of ThT over time in several wild-type strains; in all of them, the fluorescence changed when CCCP and KCl were added and showed similar variations; moreover, it was also performed in several mutant strains of some of the main K^+^ transporters (Trk1p, Trk2p, and Tok1p) of *S. cerevisiae* in order to establish the reliability of the method. Furthermore, we used different techniques, such as fluorescence in a macro and micro method (multi-well plate reader), confocal microscopy, spectrophotometry, and flow cytometry, to show the robustness and applicability of this method. These techniques have already been used in several works before [30,31,32,48], but in our case, those that were different from the accumulation experiments validated the method of obtaining a numerical value of PMP in yeast using ThT according to its accumulation.

As already reported [20], three sequential responses of fluorescence were obtained for all the different strains. First, one shows the fluorescence when ThT is added to the cells; then, an increase in fluorescence is observed after adding CCCP; finally, a decrease due to the addition of KCl, is observed, showing mainly the effectiveness of the method and that each strain generates different fluorescence intensities. Results in Figure 3 showed that the wild-type strains from different genera of yeasts reveal a similar behavior upon the addition of ThT, CCCP, and KCl; this behavior could also be observed in the results in the multi-well plates with *S. cerevisiae* Azteca (Figure 4a–c), as well as in the images obtained with the confocal microscope (Figure 5 and Figure 6), where the highest fluorescence in all the strains is presented when the CCCP is added due to the release of ThT from mitochondria [44] and then a decrease in fluorescence by the addition of K^+^ that depolarizes the membrane potential, which most likely causes the dye to be released from the cell [45].

In general, the images of the cells obtained by microscopy (Figure 5 and Figure 6) after the addition of glucose, glucose plus CCCP, and then KCl, were also consistent with the changes observed in the spectrofluorometer (Figure 3). It is important to mention that a similar tendency was observed in the plate reader results (Figure 4), but unlike the results obtained in the spectrofluorometer or by microscopy, the results obtained with this equipment were taken over a longer period of time (40 min), where agitation is also limited. Previously, the estimation of PMP or mitochondrial membrane potential with ThT in prolonged periods [32,44] had been reported, suggesting that the stability of ThT fluorescence can be maintained over time without loss of fluorescence intensity [32,49].

*S. cerevisiae* possesses three specific K^+^ transport systems encoded by *TRK1*, *TRK2*, and *TOK1* [50]. Experiments were performed in mutants of these transporters (Figure 3, Figure 4e,f and Figure 9d,e) to validate the importance of K^+^ in establishing the PMP [1]. Results with the double-mutant strain (*trk1Δ trk2Δ*), as expected, did not show the fluorescence decrease observed upon the addition of KCl because both of the transporters for the uptake of K^+^ were deleted; however, the small decrease in fluorescence when KCl is added (Figure 9d) depends on the K^+^ uptake through non-specific mechanisms [48,51,52]; the *trk2Δ* strain showed a similar result to its wild-type *S. cerevisiae* FY833 strain, as *TRK2* encodes for a low-affinity K^+^ transporter (Trk2p) [53,54,55], while the added K^+^ is taken up by the high-affinity K^+^ transporter Trk1p [56,57]; in the *trk1Δ* strain, however, depolarization was decreased due to low K^+^ uptake by the Trk2p transporter. As for the TOW mutant (lacking the *TOK1* gene, which encodes for a K^+^ outflow channel, Tok1p [58,59,60]), no significant differences were observed from its corresponding wild-type *S. cerevisiae* W303-1A—only a slight tendency towards depolarization, better seen in the non-corrected data for Figure 4f and Figure 9e, with exception on the fluorometer traces, where a gradual depolarization upon the addition of ThT was observed and a greater response to K^+^ was also obtained, differently from its genetic background (Figure 3). Moreover, an adequate result for the TOW strain was obtained in the ThT accumulation experiments. The difference in the results with this strain may be due to the sensitivity of the method and the equipment used for each experiment. 

Related also with TOW strain, Maresova et al. found that the depolarization of the plasma membrane is probably not a direct consequence of *TOK1* deletion [38]. However, when the potential decreases (for example, via K^+^ uptake, which is a natural physiological action), cells lacking the *TOK1* gene cannot react by activating Tok1p; therefore, they stay depolarized. Moreover, Maresova et al. found that the presence of Ena1-4p and Nha1p were more important for the membrane potential regulation of cells lacking the *TOK1* gene, indicating that these transporters could partially compensate for the function of Tok1p in membrane potential regeneration [38].

If a laboratory does not have all of the aforementioned instrumentation, results can be obtained either by measuring fluorescence changes in a multi-well plate reader or a confocal microscopy, as shown in Figure 4, Figure 5 and Figure 6, or by using flow cytometry (Figure 9), which also allowed us to test the hypothesis shown in Figure 7, which schematizes how ThT movements occur after the addition of different effectors such as CCCP, KCl, chitosan, and an excess of KCl.

The values of Table 4 and Table 5 show reasonable values for the plasma membrane Δ*ψ*. These values were, of course, lower when CCCP was added, which produces the efflux of the dye accumulated by the mitochondria, and even more after the addition of 10 mM KCl. However, in the case of *D. hansenii*, *M. guilliermondii*, and *R. mucilaginosa*, the correction of data could be calculated; however, when 10 mM KCl was added, no depolarization could be observed. This can be explained due to the higher hypersensitivity of the plasma membrane that these strains have for CCCP, which could be indirectly inferred from the effects of the uncoupler that, at very low concentrations, inhibited K^+^ transport (Appendix A and [41]); so, if low concentrations of CCCP not only affect the mitochondria but also the plasma membrane, the results become invalid. As has already been said, in order to be able to estimate or measure the PMP in yeasts, a much lower sensitivity of the plasma membrane compared to the mitochondria is required for the effectiveness of the method. Moreover, these three yeasts are defined as oleaginous [21,61,62], meaning that they are able to produce and store lipids at a higher rate than other yeasts. Considering that ThT is a cationic hydrophobic dye whose partition coefficient has been calculated to be less hydrophobic than cyanine (DiSC_3_(3)), but higher than Rhodamine 6 G, for example [30], the rate at which it is bonded to lipids inside these kinds of cells could make it difficult for a correction to be made. 

Non-corrected values obtained with the mutants of the K^+^ transporters, as expected, were clear for the *trk1Δ* mutant and the double *trk1Δ trk2Δ* mutant (Table 5 and Figure 8). Unfortunately, as already mentioned, the corrected values upon the addition of KCl for their corresponding wild-type strain and *trk2Δ* mutant gave invalid values.

In summary, although the corrected data were not suitable for all strains, the methods, following the fluorescence changes of ThT by various means, are adequate to estimate the PMP changes of the different strains tested. The results obtained by measuring the non-corrected uptake of the dye are the best options existing to date. The use of mutant strains of K^+^ transporters was important to validate and show the veracity of our method, and since, as reported by Rodriguez-Navarro in 2000, each of these transport systems (Trk1p, Trk2p, and Tok1p) share sequence homology with K^+^ transporters present in bacteria, plant, animal, and other fungal cells [63], we can suggest that this method could be used in other cell types. 

The other method used to prove that the changes obtained with ThT were accurate was flow cytometry, which has the ability to measure the optical and fluorescence characteristics of different kinds of cells, from mammalian cells to microorganisms (in this case, yeasts). The fluorescence emission derived from a fluorescence probe is proportional to the amount of fluorescent molecules bound to the cell or cellular component [64]; therefore, it can also be used to estimate the changes of the PMP of the strains used following the changes in fluorescence. Madrid et al. already observed fluorescence changes with flow cytometry in cells stained with DiOC_6_(3) to estimate the PMP [48].

With all strains, an increase in fluorescence was observed upon the addition of the dye, which was larger after adding CCCP, and decreased after the addition of KCl. Generally, flow cytometry results for the wild-type and mutant strains confirmed what was observed with other techniques (Figure 9c–e). This method may be another alternative to following the relative fluorescence changes under the tested conditions.

Overall, changes in PMP in conventional, non-conventional, and K^+^ transport mutant yeasts could be monitored via ThT fluorescence, and these depend on the method used, in which a similar pattern of change could be observed. However, some of the results observed—for example, in the case of *D. hansenii*—can be explained by known sensitivity to the presence of protonophores such as CCCP [41,65], which could cause fluorescence changes to be minimal compared to other wild-types. In the case of *M. guilliermondii* (Figure 4d, Figure 6 and Figure 9c), a yeast known to produce and accumulate riboflavin [66], which has two excitation wavelengths (350 nm and 450 nm) and emits at 530 nm [67,68], those wavelengths are very close to that of ThT, making the measurement of the ThT fluorescence and absorbance in accumulation experiments difficult. An additional problem in tracking ThT fluorescence changes was that with *R. mucilaginosa*, the fluorescence changes were lower (Figure 4d, Figure 6 and Figure 9c, probably caused by its red color and carotenoid accumulation, where the absorption, excitation, and emission peaks of their pigments are close to the ones of ThT [69,70]. These issues probably caused *D. hansenii, M. guilliermondii*, and *R. mucilaginosa* to show very little changes in fluorescence after the addition of CCCP or KCl in methodologies such as plate reader (Figure 4d), confocal microscopy (Figure 6), and flow cytometry (Figure 9c).

On the other hand, although ThT is so far the most convenient dye, to estimate the changes of the PMP by following its fluorescence, other dyes have been shown to function similarly, and studies with some of them were found to provide similar results [20,30]; García-Navarrete also found that ThT was ideal for the long-term microfluidics using yeast cells to estimate their PMP [32].

Overall, we showed that the use of different techniques to determine the fluorescence changes of ThT works as expected; even though we cannot always observe a statistical difference, we can observe a pattern that remains in all strains tested. 

## 5. Conclusions

In our work, using various methodologies and comparative analysis, we demonstrate that following the fluorescence changes and accumulation of ThT, this dye can be used to qualitatively and quantitative obtain actual PMP values; although not adequate for all strains and their mutants, with the different methods, we were able to estimate the PMP changes in most of the different strains tested.

The validity of the method in estimating the numeric value of PMP is confirmed, as shown by the data obtained with different strains, mainly with K^+^ transport mutants. 

It is possible that these methods could be applied to other yeast strains or cell lines, as suggested in the literature, always bearing in mind the concentration to be used for uncoupling. It is also a method that can be carried out via different methodologies depending on equipment availability. As to the results obtained by measuring the non-corrected uptake of the dye, this is the best option existing to date.

## Figures and Tables

**Figure 1 jof-09-00948-f001:**
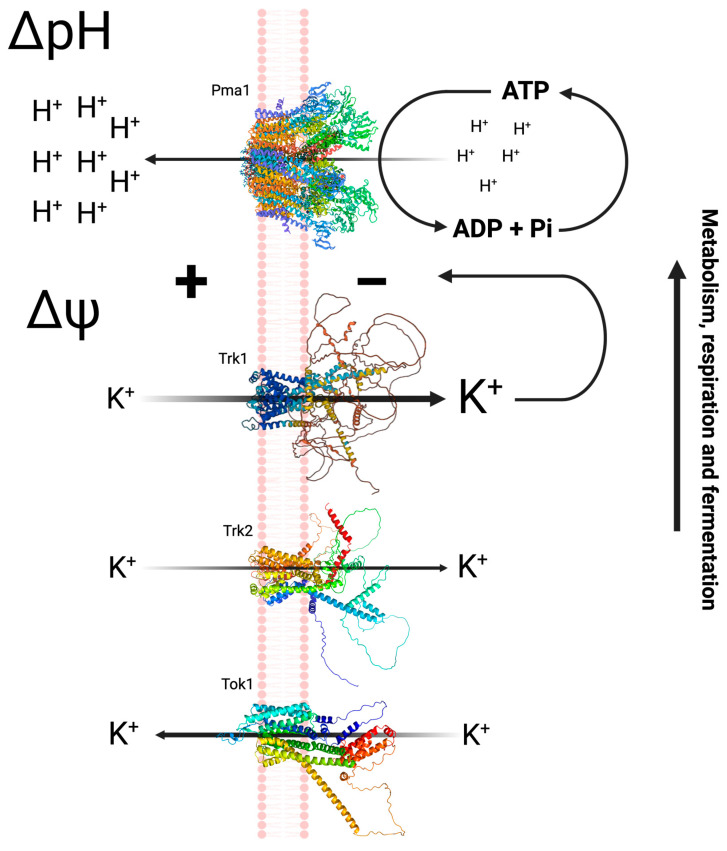
Mechanism of acidification and K^+^ transport in *S. cerevisiae*. Pma1p acidifies the external medium by pumping out protons, generating a pH difference with the interior of the cell and an electric potential difference (Δ*ψ*) that is used to internalize K^+^ ions by Trk1p (a high affinity transporter) and Trk2p (a low affinity transporter). The tridimensional structure of Pma1p (PDB: 7VH5) originally from [4], Trk1p was obtained from the UniProt database (P12685) [5], and the structure of Trk2p and Tok1p was modeled using AlphaFold [6,7]. Created with BioRender.com.

**Figure 2 jof-09-00948-f002:**
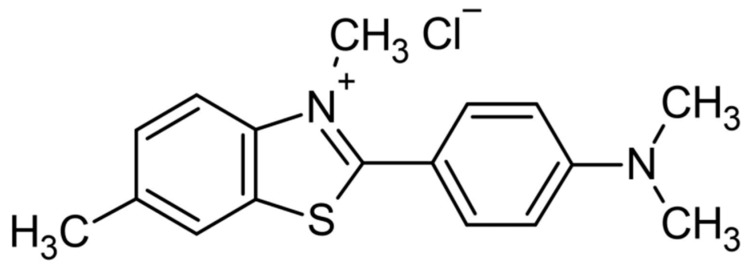
Chemical structure of thioflavin T (ThT).

**Figure 3 jof-09-00948-f003:**
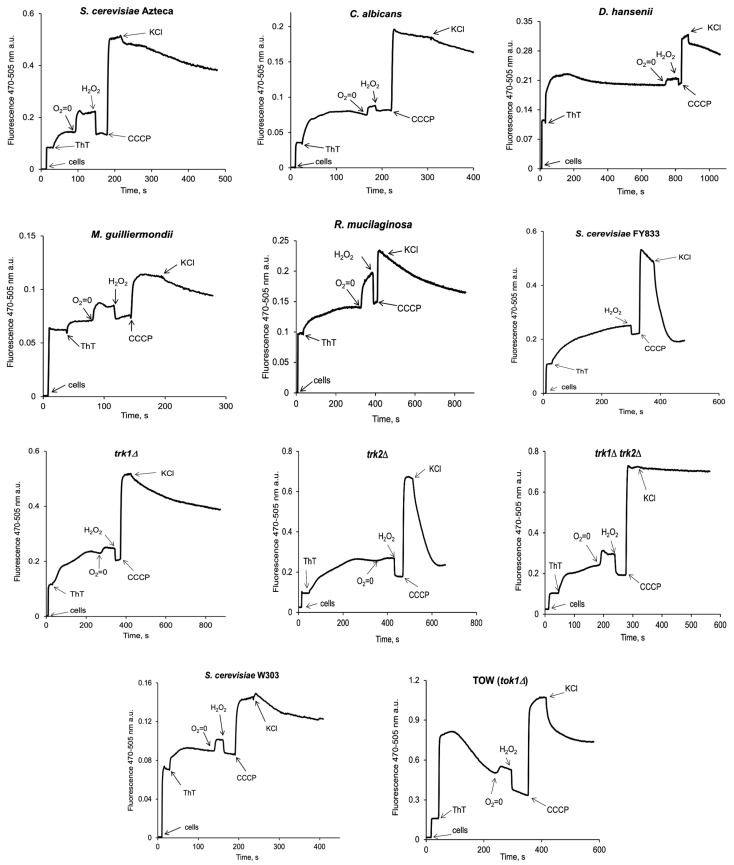
Estimation of the PMP via the fluorescence changes of ThT in wild-type strains of different yeast genera and mutants of the main K^+^ transporters in *S. cerevisiae*. The incubation medium contained 10 mM MES-TEA buffer, 10 µM BaCl_2_, and 20 mM glucose. After the addition of 15 µM ThT, the cells (50 mg, w. w.) were added. After variable time, a small increase in fluorescence was observed (O_2_ = 0, indicating the point where oxygen is depleted), which decreased upon the addition of 5 µL of 3% H_2_O_2_ (as an oxygen source). Then, adding 10 µM CCCP produced a large increase in fluorescence, and after the addition of 10 mM KCl, in most tracings, a variable decrease was observed. Fluorescence changes were followed at 470–505 nm, excitation, and emission wavelengths, respectively, in an Olis-modified SLM spectrofluorometer with constant magnetic stirring and 30 °C. Representative traces from three biological replicates are shown.

**Figure 4 jof-09-00948-f004:**
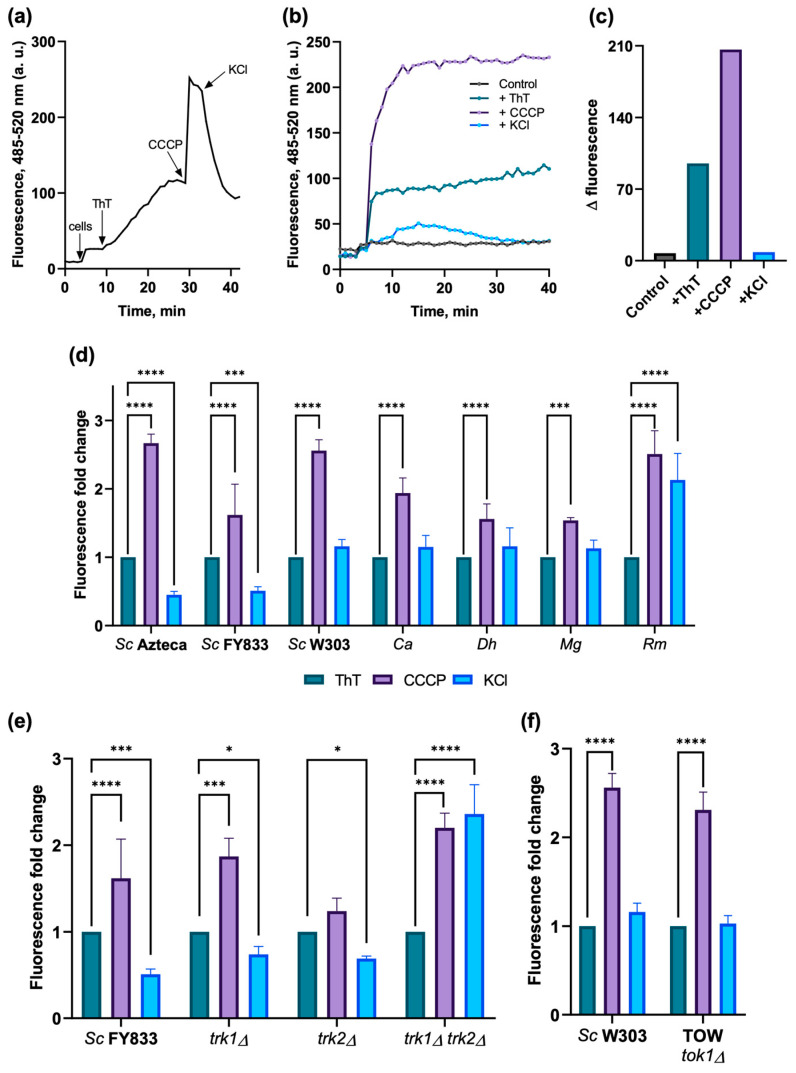
PMP qualitative estimation via fluorescence of ThT measured in a multi-well plate reader. In (**a**), a time-course tracing was performed in 10 mM MES-TEA buffer, pH 6.0, 20 mM glucose, 10 µM BaCl_2,_, adding 5 mg cells (w. w.), 15 µM ThT, then 10 µM CCCP, and finally 10 mM KCl. (**b**) Single traces under each condition. (**c**) The Δ of fluorescence obtained from (**b**). In (**d**), a value of 1.0 was defined for the Δ of fluorescence emitted by the sample with only ThT and yeast, since each strain displayed different values. In (**e**,**f**), this normalization was made only for the wild-type strains. Subsequently, the relative fluorescence of the samples with CCCP and CCCP + KCl, were reported as a fold increase value regarding the ThT cell sample (*n* = 3–5). Two-way ANOVA with a Dunnett’s multiple comparisons tests were performed for statistics with *p*-values of 0.05, (*) 0.0332; (***) 0.0002; (****) 0.0001. Representative results from three-to-five similar experiments are shown. *Sc*: *Saccharomyces cerevisiae*; *Ca*: *Candida albicans*; *Dh*: *Debaryomyces hansenii*; *Mg*: *Meyerozyma guilliermondii*; *Rm*: *Rhodotorula mucilaginosa*.

**Figure 5 jof-09-00948-f005:**
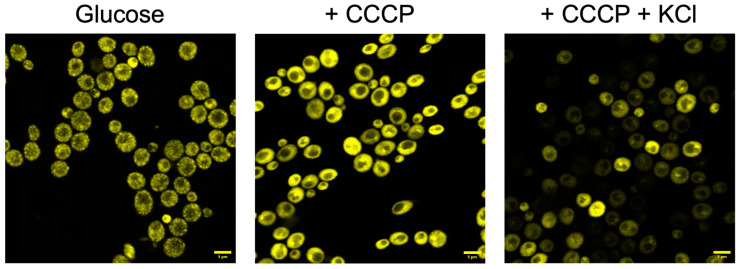
Images of S. cerevisiae (Azteca strain) obtained by incubating the cells with 20 mM glucose, 10 µM BaCl_2_, and 15 µM ThT in 10 mM MES-TEA buffer, pH 6.0, after adding 10 µM CCCP and after the further addition of 10 mM KCl. Images were taken as described in the Section 2. The scale bar corresponds to 5 µm. Representative results from three similar experiments are shown.

**Figure 6 jof-09-00948-f006:**
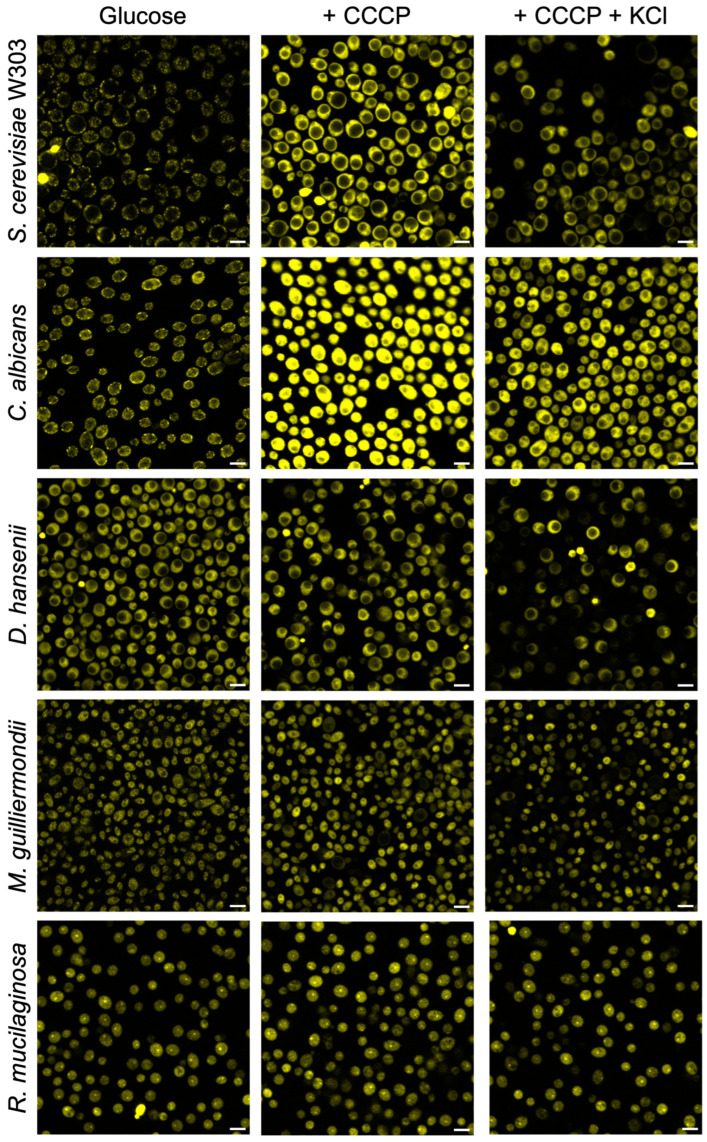
Fluorescence images of the different strains studied upon the incubation with ThT and its behavior with CCCP and the further addition of KCl. The cells, 25 mg (w. w.), were added to a mixture of 10 mM MES-TEA buffer, pH 6.0, 10 µM BaCl_2_, 20 mM glucose, and 15 µM ThT. In the same mixture, 10 µM CCCP and after, 10 mM KCl, were added. Microscopy images were obtained as described in the Section 2. The scale bar corresponds to 5 µm. *n* = 25–60 of four biological replicates for each strain.

**Figure 7 jof-09-00948-f007:**
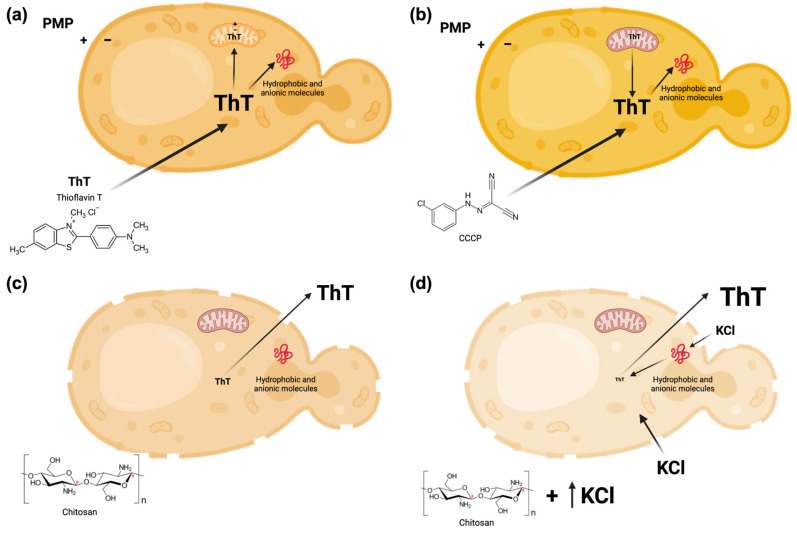
Schematic representation of the distribution of ThT or any other cationic and hydrophobic molecule used to estimate the PMP in yeast. Created with BioRender.com. Refer to the text for further explanation.

**Figure 8 jof-09-00948-f008:**
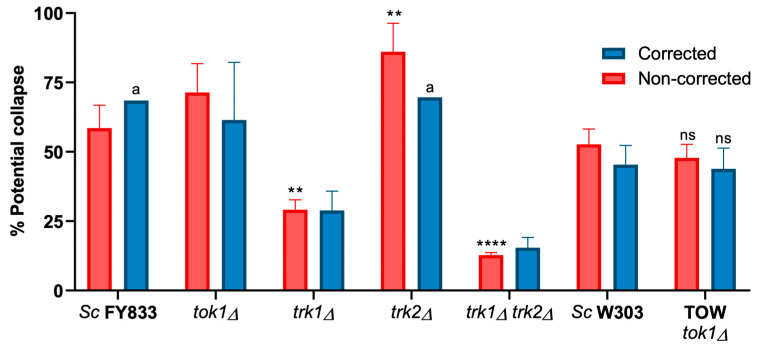
Percentage depolarization by the addition of KCl. Values calculated from the results of Table 4, *n* = 2–5. Corrected values were obtained by the addition of chitosan and KCl excess, as defined for Table 3 (see the Section 2). The first statistical analysis was made by comparing the wild-type *Sc* FY833 and its mutants in the non-corrected data with one-way ANOVA and Dunnett’s multiple comparisons tests with a *p*-value of 0.05, (**) 0.0021; (****) 0.0001; and a second analysis was made comparing the wild-type *Sc* W303-1A and TOW with Student-*t* test with a *p*-value of 0.05, (ns: non-significative) 0.1234. (a) Data obtained from Table 4; in these strains, the corrected data of only one experiment could be calculated; therefore, neither mean not SD were determined.

**Figure 9 jof-09-00948-f009:**
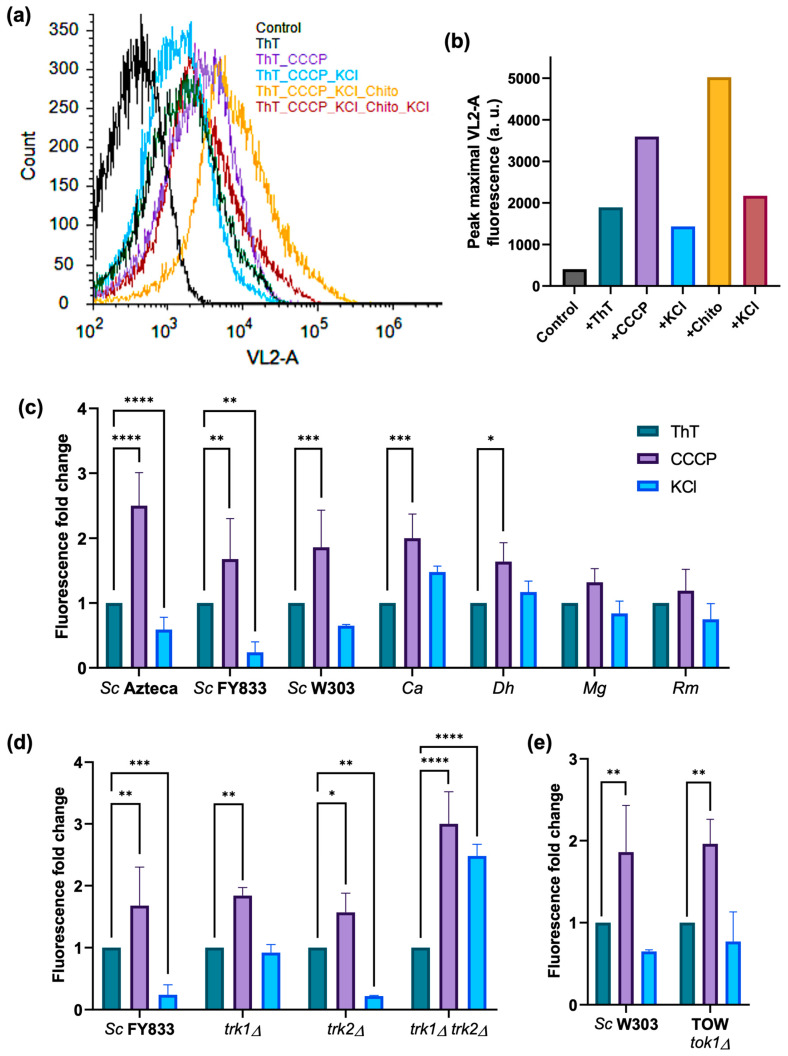
The use of flow cytometry to estimate the changes in PMP by the fluorescence of ThT. Histograms and graphs were obtained by measuring the ThT fluorescence in a flow cytometer as described in the Section 2. In (**a**), the histograms of each experiment were plotted together: the first one (black line), shows the control of cells without ThT; then, the successive additions of 15 µM ThT, 10 µM CCCP, and 10 mM KCl, then 50 µM chitosan, and finally, 200 mM KCl in the same incubation buffer. From 3 to 5 min of incubation for each sample were enough to start the acquisition of data in channel VL2-A. (**b**) Peaks of the mean maximal fluorescence obtained in (**a**). In (**c**), a value of 1.0 was defined for the fluorescence emitted by the sample with only ThT and yeast, since each strain displayed different values. Subsequently, the relative fluorescence of the samples with CCCP and CCCP + KCl were reported as a fold increase value compared to the ThT sample. In (**d**,**e**), a value of 1.0 was set for the fluorescence of the wild-type strains only. Two-way ANOVA and a Dunnett’s multiple comparisons test were performed for statistics with a *p*-value of 0.05, (*) 0.0332; (**) 0.0021; (***) 0.0002; (****) 0.0001. Representative results from three-to-five similar experiments are shown.

**Table 1 jof-09-00948-t001:** Strains used in this work.

Strain	Genotype	Source
*Candida albicans*		ATCC 10231
*Debaryomyces hansenii* Y7426		US Department of Agriculture, Peoria, IL,
*Meyerozyma guilliermondii*		[33]
*Rhodotorula mucilaginosa*		ATCC 66034
*Saccharomyces cerevisiae* Azteca		Commercial strain
*Saccharomyces cerevisiae* FY833	*MATa (his3Δ200 leu2Δ1 lys2Δ202 trp1Δ63 ura3-52)*	[34]
*trk1Δ*	*MATa (his3Δ200 leu2Δ1 lys2Δ202 trp1Δ63 ura3-52) trk1Δ::kanMX*	[35]
*trk2Δ*	*MATa (his3Δ200 leu2Δ1 lys2Δ202 trp1Δ63 ura3-52) trk2Δ::HIS3*	[35]
*trk1Δ trk2Δ*	*MATa (his3Δ200 leu2Δ1 lys2Δ202 trp1Δ63 ura3-52) trk1Δ::TRP1-trk2Δ::kanMX*	[36]
*Saccharomyces cerevisiae* W303-1A	*MATa (ade2-1 can1-100 his3-11,15 leu2-3,112 trp1-1 ura3-1 mal10)*	[37]
TOW (*tok1Δ*)	*MATa (ade2-1 can1-100 his3-11,15 leu2-3,112 trp1-1 ura3-1 mal 10) tok1Δ::kanMX*	[38]

**Table 2 jof-09-00948-t002:** Values of the internal cytoplasmic water volume of the different strains used.

	% Dry Weight	Internal Cell Volume mL/g (w. w.)	Vacuole/Cell Volume Ratio	Total Vacuole Water/g Cells (mL)	Cytoplasm Volume/g Cells (mL)	Cytoplasm Volume/150 mg Cells (mL)
*S. cerevisiae* Azteca	22.2	0.430	0.117	0.050	0.380	0.0569
*S. cerevisiae* FY833	18.0	0.424	0.276	0.117	0.307	0.0460
*S. cerevisiae* W303-1A	22.4	0.440	0.252	0.111	0.329	0.0494
*C. albicans*	27.0	0.410	0.150	0.061	0.349	0.0523
*D. hansenii*	21.0	0.440	0.091	0.040	0.400	0.0600
*M. guilliermondii*	28.2	0.390	0.099	0.039	0.351	0.0527
*R. mucilaginosa*	14.5	0.420	0.071	0.030	0.390	0.0585
*trk1Δ*	19.9	0.460	0.124	0.057	0.403	0.0605
*trk2Δ*	18.3	0.478	0.265	0.127	0.352	0.0528
*trk1Δ trk2Δ*	18.4	0.463	0.142	0.066	0.397	0.0595
TOW	15.4	0.529	0.264	0.140	0.390	0.0585

Dry weight was measured in 100 µL yeast suspension (50 mg, w. w.), drying at 95 °C in aluminum trays for 24 h, transferring them to a desiccator, and weighing them in an analytical balance. To determine the internal cell volume, the distribution of ^14^C-Inulin was measured by mixing 0.5 mL of cells (0.25 g, w. w.) with 0.015 mL of ^14^C-inulin in ice cold water, centrifuging the cells, and measuring the radioactivity, as described in the Section 2. The external water volume was obtained, which, subtracted from the total volume, gave the total cell volume. The dry weight value of an equal amount of the cells was also subtracted to obtain the total cell water volume as in [41]. Then, since the dye does not enter the vacuole, the volume of this organelle was subtracted from the total cell water volume to obtain the volume of the cytoplasm only, where the dye actually accumulates. The values obtained for each strain were then calculated for 150 mg of cells used in the dye accumulation tests. For dry weight measurements, at least three biological replicates with 8–10 technical repetitions each were made. To determine the internal cell volume with ^14^C-Inulin, three biological replicates with six samples each were included, and the vacuole/cell volume ratio was calculated from 60 to 300 cells of three–four different yeasts cultures. Average values are reported.

**Table 3 jof-09-00948-t003:** Values of Δ*ψ* for the different wild-type strains obtained as described.

	Non-Corrected	Corrected
	Δ*ψ* (mV) ± SD	Δ*ψ* (mV) ± SD
***S.c.* Azteca**		
Glucose	−201.6 ± 3.4	−192.8 ± 22.3
CCCP	−166.2 ± 9.9	−148.3 ± 13.9
CCCP + KCl	−100.5 ± 6.1	−72.5 ± 8.0
***S. c*. FY833**		
Glucose	−164.3 ± 12.6 ^b^	−153.51 ± 12.3 ^b^
CCCP	−130.4 ± 11.1	−115.46 ± 12.8
CCCP + KCl	−36.9 ± 15.2	−71.66 *
***S.c.* W303-1A**		
Glucose	−179.3 ± 15.6 ^b^	−160.7 ± 16.5 ^b^
CCCP	−154.3 ± 16.4	−133.1 ± 15.8
CCCP + KCl	−112.1 ± 24.1	−96.2 ± 27.7
** *C. albicans* **		
Glucose	−248.8 ± 14.9	−228.2 ± 19.0
CCCP	−185.7 ± 16.5	−153.2 ± 15.5
CCCP + KCl	−142.0 ± 18.1	−140.4 ± 19.9
** *D. hansenii* **		
Glucose	−154.4 ± 12.1 ^a,b^	−116.9 ± 17.1 ^a,b^
CCCP	−120.7 ± 12.2	−56.7 ± 9.7
CCCP + KCl	−101.5 ± 10.6	−73.3 ± 18.4
** *M. guilliermondii* **		
Glucose	−167.1 ± 13.1 ^b^	−135.7 ± 8.6 ^a,b^
CCCP	−135.2 ± 10.9	−96.9 ± 12.6
CCCP + KCl	−111.8 ± 13.5	−115.5 ± 19.0
** *R. mucilaginosa* **		
Glucose	−148.3 ± 20.3 ^a,b^	−139.0 ± 13.8 ^a,b^
CCCP	−70.5 ± 10.8	−32.4 ± 2.6
CCCP + KCl	−75.1 ± 8.3	−72.7 ± 8.8

Cells, 150 mg (w. w.), were incubated in 3.0 mL final volume of the following mixture to obtain the first values (glucose): 10 mM BaCl_2_, 10 mM glucose, and 100 µM ThT. Then, 10 µM CCCP was added, followed by 10 mM KCl (non-corrected). For the corrected values, 100 µg of chitosan was also added to obtain the values of bound dye due to its hydrophobicity and cationic nature. In another tube, the same amount of chitosan was added, followed, after 10 min, by 200 mM KCl to obtain, by difference, the value of the dye bound only by its hydrophobic nature. Data were obtained and calculated as described in the Section 2 and reported as average ± SD of *n* = 4–13. * For FY833, five biological replicates and two technical replicates were conducted; however, when correcting the data, only a single experiment could be calculated. Two-way ANOVA with a Tukey’s multiple comparisons tests were performed for corrected and non-corrected values, with a *p*-value of 0.05: (a) statistical difference with *S. cerevisiae* Azteca; (b) statistical difference with *C. albicans*.

**Table 4 jof-09-00948-t004:** Values of Δ*ψ* for *S. cerevisiae* FY833 and W303-1A and its mutants.

	Non-Corrected	Corrected
	Δ*ψ* (mV) ± SD	Δ*ψ* (mV) ± SD
**FY833**		
Glucose	−164.3 ± 12.6	−153.51 ± 12.3
CCCP	−130.4 ± 11.1	−115.46 ± 12.8
CCCP + KCl	−36.9 ± 15.2	−71.66 *
** *trk1Δ* **		
Glucose	−186.4 ± 5.7 ^a^	−174.9 ± 4.8 ^a^
CCCP	−143.1 ± 7.3	−128.1 ± 8.3
CCCP + KCl	−134.9 ± 4.8	−126.6 ± 8.3
** *trk2Δ* **		
Glucose	−157.0 ± 8.6 ^ns^	−147.9 ± 7.3 ^ns^
CCCP	−129.9 ± 4.2	−118.8 ± 5.5
CCCP + KCl	−54.4 ± 4.9	−21.17 **
** *trk1Δ trk2Δ* **		
Glucose	−193.8 ± 12.7 ^a^	−179.7 ± 13.9 ^a^
CCCP	−158.9 ± 10.6	−143.2 ± 11.8
CCCP + KCl	−163.8 ± 9.7	−155.9 ± 9.2
**W303-1A**		
Glucose	−186.8 ± 7.0	−167.7 ± 11.8
CCCP	−162.9 ± 4.4	−143.8 ± 6.4
CCCP + KCl	−117.9 ± 21.7	−121.2 ± 3.5
**TOW**		
Glucose	−172.8 ± 9.4 ^ns^	−159.4 ± 6.7 ^ns^
CCCP	−141.1 ± 7.3	−125.1 ± 4.5
CCCP + KCl	−96.6 ± 9.3	−82.1 ± 9.5

Values were obtained as described in Table 3 and calculated as described in the Section 2 and reported as average ± SD of *n* = 4–13 * For FY833, five biological replicates and two technical replicates were conducted; however, when correcting the data, only a single experiment could be calculated. ** For *trk2Δ*, four biological replicates and two technical replicates were conducted; however, when corrected the data, only a single experiment could be calculated. The first statistical analysis was performed, comparing the wild-type *Sc* FY833 and its mutants in the non-corrected and corrected data via two-way ANOVA and Dunnett’s multiple comparisons tests with a *p*-value of 0.05; (a) statistical difference with *Sc* FY833; and a second analysis was conducted comparing the wild-type *Sc* W303-1A and TOW via two-way ANOVA and Sidak’s multiple comparisons tests with a *p*-value of 0.05. ns, non-statistical difference.

**Table 5 jof-09-00948-t005:** Values obtained for the PMP adding 10 or 5 µM CCCP of *M. guilliermondii*.

	Glucose	CCCP	CCCP + KCl
10 µM CCCP	−167.1	−135.2	−111.8
5 µM CCCP	−121.81	−113.9	−97.4

Experiment performed as described in Table 3.

## Data Availability

Not applicable.

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
