# Peer review of "The Use of Thioflavin T for the Estimation and Measurement of the Plasma Membrane Electric Potential Difference in Different Yeast Strains"

_jof, 2023, doi:10.3390/jof9090948_

Round 1
Reviewer 1 Report
In this manuscript, Pena et al. described a novel method to quantify the membrane potential of budding yeast cells. They chose thioflavin (ThT), a cationic dye that has been employed in their own earlier studies (Calahorra et al. 2019 Journal of Bioenergetics and Biomembranes). Using ThT, the authors measured the membrane potential of several species of yeasts as well as a few budding potassium channel mutants. They applied three methods, spectrofluorometer, fluorescence microscopy and flow cytometry to measure the fluorescence of ThT in yeast cells. To validate this probe, they applied different types of stress, including CCCP, hydrogen peroxide and KCl, to alter the membrane potential. Overall, the authors demonstrated that ThT could be a potentially useful tool for measuring membrane potential of yeast cells.
The study examines an important question of yeast physiology in quantitative studies of membrane potential. There is no widely used tool available for measuring membrane potential of yeast cells. This study thus shall be of interest to yeast cell biologists.
The only major concern is that the validity of this fluorescence dye-based membrane potential probe remains unclear even after this study. The authors’ group remains the only one lab that has employed this method. To establish the wide applicability of this probe, the reviewer recommends that the authors compare theirs with the commercially available dye for probing membrane potentials such as FluoVolt (ThermoFisher). In addition, the authors shall provide a full mechanistic explanation of how the fluorescence of ThT correlates with the membrane potential.
Minor concerns,
1. Fig. 3: Please label which one is Pma1 and which one is Trk1. And please add Tok1 and Trk2 in this model.
2. Fig. 3: It is unclear what “O2=0” means and how it was determined. Please clarify.
3. Fig. 3: The subpanels are too big. Please resize the subpanels so that all of them can fit into one page.
4. Fig. 3: It is misleading that each panel uses its own scale on both X and Y axis even though all the experiments were carried out with the same method. Please use the same scales on X (0-1000s) and Y (0-0.8) axis for all the plots.
5. Fig. 4: it is confusing that excitation and emission spectrum (485/520) of this plate reader were different from that of the spectrofluorometer (470/505), even though the same dye ThT was used. Please clarify the reason why different spectrums were used. Similarly, please clarify the choice of using 458/463-550 as excitation and emission spectrum when imaging the yeast cells with fluorescence microscopy. Lastly how do those parameters compare to the actual peak excitation/emission spectrum of the dye ThT?
6. Fig. 4: It remains unclear whether there is any significant difference in the membrane potential of the various yeast strains (Fig. 4d and 4e). Please provide statistical tests comparing the membrane potential among the yeasts.
7. Fig. 5 and 6: it is confusing when these micrographs were taken after addition of either CCCP or KCl. Please clarify.
8. Fig. 5 and 6: missing scale bars
9. Fig. 7: this is not essential to the paper. Please move it to the supplemental materials.
10. Fig. 8: Several columns are missing error bars. Please add those. The last column (TOW null) is also missing “not significant” label. Please add it.
11. Fig. 8: it is confusing what “corrected” and “not corrected” mean. Please clarify in the figure legends.
12. Fig. 9: panel a is confusing. Please clarify what the y-axis “count” and what the x-axis “VL2-A” is.
13. Fig. 9: Similar to Fig. 4, it is unclear whether there is difference among the various yeasts and the mutants. Please provide statistical tests for 9C and 9d to determine whether the membrane potential of different yeasts is different.
14. Please discuss why the result in Fig. 9d differs from that shown in Fig. 4e, even though both used ThT to measure membrane potential.
Overall, the manuscript provides an interesting method of measuring yeast membrane potential, although the method is not entirely new. The reviewer would recommend the recommended revisions before acceptance.
Acceptable
Author Response
REVIEWER 1
Comments and Suggestions for Authors
In this manuscript, Pena et al. described a novel method to quantify the membrane potential of budding yeast cells. They chose thioflavin (ThT), a cationic dye that has been employed in their own earlier studies (Calahorra et al. 2019 Journal of Bioenergetics and Biomembranes). Using ThT, the authors measured the membrane potential of several species of yeasts as well as a few budding potassium channel mutants. They applied three methods, spectrofluorometer, fluorescence microscopy and flow cytometry to measure the fluorescence of ThT in yeast cells. To validate this probe, they applied different types of stress, including CCCP, hydrogen peroxide and KCl, to alter the membrane potential. Overall, the authors demonstrated that ThT could be a potentially useful tool for measuring membrane potential of yeast cells.
The study examines an important question of yeast physiology in quantitative studies of membrane potential. There is no widely used tool available for measuring membrane potential of yeast cells. This study thus shall be of interest to yeast cell biologists.
- The only major concern is that the validity of this fluorescence dye-based membrane potential probe remains unclear even after this study. The authors’ group remains the only one lab that has employed this method. To establish the wide applicability of this probe, the reviewer recommends that the authors compare theirs with the commercially available dye for probing membrane potentials such as FluoVolt (ThermoFisher). In addition, the authors shall provide a full mechanistic explanation of how the fluorescence of ThT correlates with the membrane potential.
Answer:
In our manuscript, we mention that one group, besides us, has used this dye with satisfactory results; we feel honored that our proposal has started to be used [references 1-4 of this response to reviewers].
Fluovolt has been used to measure the membrane potential of animal cells, and it is highly hydrophobic. In this respect, our results are consistent with those obtained with another dye, DiSC3(3) [5]; this dye gives similar fluorescence changes to those of ThT; however, it is more hydrophobic, and is accumulated by the cells to a much larger extent, due to its more hydrophobic nature. But fluorescence results obtained with that dye are not very different to those obtained with ThT. In fact, we have reported that similar, although not accurate values of the PMP can be obtained by measuring the accumulation of several dyes in baker’s yeast [6]. Regarding what we consider is the main characteristic of ThT that makes it a better choice, is its hydrophobicity, which is about one third that of DiSC3(3) [6]. Given that Fluovolt stays inserted at the membrane, its mechanism of action would surely be different to what we are proposing for ThT. We had 10 days as the deadline for a Major revision and obtaining a chemical reactive in a Third World Country as ours can take as much as a month. This is certainly much interesting to address, but it would have to be explored in the future. We apologize. A complete list of the dyes we have compared is in references 6 and 7 of this response.
Regarding the mechanism, we consider having made clear that ThT accumulates due to its positive charge and interacts with the internal components of the cell, which produces an increase of its fluorescence, but when it accumulates in excess by the mitochondria, a decrease is observed. In fact, this phenomenon is well known. As it happens with other dyes, the fluorescence increases when they are mixed with proteins, such as albumin, or different fatty acids, among other molecules. We showed long ago that simply mixing DiSC3(3) with a yeast extract greatly increases its fluorescence [5], and this has been shown with other dyes as well.
(Please look at the attached file where there are some figures in the response)
Gitler at al. [8] introduced for the first time the use of ethidium bromide to estimate the membrane potential of mitochondria, by following its fluorescence. What they found was a decrease of fluorescence as the membrane potential increased. Later, by using yeast mitochondria, we also found a decrease of fluorescence when the membrane potential increased, and vice versa [9]:
- Fig. 3: Please label which one is Pma1 and which one is Trk1. And please add Tok1 and Trk2 in this model.
Answer:
We already made the changes in the figure, we appreciate your suggestion.
- Fig. 3: It is unclear what “O2=0” means and how it was determined. Please clarify.
Answer:
At that point oxygen is depleted, mitochondria release part of the dye and a jump in fluorescence is seen. The addition of a low amount of H2O2 as an oxygen source allows mitochondria to be energized again and fluorescence returns to its original level before the depletion of oxygen. This is also indicated in the text, where we mention that around one minute after the addition of ThT the cells reached anaerobiosis, and this time is different for each strain as it is now described in the text and also making reference to Fig. S3. The first study where we observed this response was in 1984, with DiSC3(3) [5]. Anyway, we add the information of the label at the figure legend so it can be more easily understood.
- Fig. 3: The subpanels are too big. Please resize the subpanels so that all of them can fit into one page.
Answer:
We already addressed this change suggested by the reviewer.
- Fig. 3: It is misleading that each panel uses its own scale on both X and Y axis even though all the experiments were carried out with the same method. Please use the same scales on the X (0-1000s) and Y (0-0.8) axis for all the plots.
Answer:
It is important to note that each strain has a different response level to the addition of ThT, CCCP, KCl, which leads to a different level of fluorescence that could be measured. Moreover, the time response is also not comparable between each strain, so if we scale all the strokes we would not be able to detect the differences in fluorescence in the strains that exhibit small changes.
- Fig. 4: it is confusing that the excitation and emission spectrum (485/520) of this plate reader were different from that of the spectrofluorometer (470/505), even though the same dye ThT was used. Please clarify the reason why different spectrums were used. Similarly, please clarify the choice of using 458/463-550 as excitation and emission spectrum when imaging the yeast cells with fluorescence microscopy. Lastly, how do those parameters compare to the actual peak excitation/emission spectrum of the dye ThT?
Answer:
In the literature and in the different supplier’s web sites, we can observe that the excitation spectrum of ThT is around 385 nm to 450 nm and the emission spectrum is around 445 nm to 482 nm (for example, https://biotium.com/product/thioflavin-t-high-purity-grade/). It has also been reported that the wavelength of ThT changes according to the polarity of the solvent in which it is dissolved [10]. Moreover, we found out that the addition of cells and CCCP can displace the wavelength to higher values, so when started to work with this dye (as with any dye we have worked with), we made spectrograms to determine the wavelength of excitation and emission with and without cells and CCCP in the fluorometer.
An example of the spectrograms obtained is in the attached file.
(Please look at the attached file where there are some figures in the response)
Unlike the spectrofluorometer, which has monochromators to precisely select the wavelengths at which we were about to work; other equipment such as the plate reader or microscope, have filters that include a set of wavelengths close to those obtained in the spectrofluorometer. We selected those filters that were close enough to that obtained in the fluorometer with reliable results.
- Fig. 4: It remains unclear whether there is any significant difference in the membrane potential of the various yeast strains (Fig. 4d and 4e). Please provide statistical tests comparing the membrane potential among the yeasts.
Answer:
In figure 4 we show the changes in the fluorescence fold change that ThT exhibits in each strain due to the addition of different effectors (CCCP and KCl), the values obtained are normalized to 1 with the fluorescence emitted by cells mixed with ThT.
These values indicate only a trend of change in fluorescence and allow us to know if a cell is depolarized or not. Since these are normalized values, it is impossible to perform a multiple statistical analysis.
The actual numerical values of PMP can be seen in Tables 3 and 4, where through the accumulation of ThT it was possible to obtain a numerical value of PMP, and to see how different the values are in the different strains.
Following your suggestion, we performed statistical analysis for Tables 3 and 4 to compare the different wild type yeasts and the mutants with their corresponding wild-types. We also added a comment in the text, regarding what was observed.
- Fig. 5 and 6: it is confusing when these micrographs were taken after addition of either CCCP or KCl. Please clarify.
Answer:
In section 2.4 of Materials and methods, and in section 3.3 of Results, we explained that all the micrographs were taken with glucose and after the addition of CCCP or CCCP plus KCl. This is also included in the figures caption.
- Fig. 5 and 6: missing scale bars
Answer:
We already added a scale bar in all the figures and we referred to it in the figures caption. This was done from the original calibrated .lsm files. We apologize for missing this basic detail and thank the reviewer for the suggestion.
- Fig. 7: This is not essential to the paper. Please move it to the supplemental materials.
Answer:
We included Fig. 7 just in that part of the article so what we are describing would be better understandable in a graphical way. Please allow us to keep it in that way so the readers do not have to bother looking for it in the supplementary material.
- Fig. 8: Several columns are missing error bars. Please add those. The last column (TOW null) is also missing “not significant” label. Please add it.
Answer:
For strains Sc FY833 and its mutant trk2Δ it was impossible to determine the percentage of PMP collapse with the corrected data, corresponding to the CCCP + KCl condition, due to the fact that for the strain Sc FY833 (5 biological replicates and two technical replicates each, 10 data) and the trk2Δ mutant (4 biological replicates and two technical replicates, 8 data), only one corrected value could be obtained, for this reason no error bars are presented. We had reported it only as a hyphen in Table 4, from which the calculations were made, however, we included the value, now presented without standard deviation, making the appropriate clarification in the table footer with an asterisk. This is the reason why no statistics were performed for this series of data, but only for the non-corrected data.
The label "not significant" has already been added for TOW column compared to Sc W303.
- Fig. 8: it is confusing what “corrected” and “not corrected” mean. Please clarify in the figure legends.
Answer:
In section 3.4 of Materials and methods, and Figure 7, we mention that the corrected values were obtained when we added chitosan and an excess of KCl (200mM). In this case the chitosan permeabilize the cell, causing the loss of potential and the excess of KCl displace the ThT bound to anionic components in the cell such as proteins. That lets us estimate the real amount of ThT that is transported inside the cell and is not taken outside after the first addition of KCl (20mM).
This is now clarified in the tables footer and in the figure 8 legend.
- Fig. 9: panel a is confusing. Please clarify what the y-axis “count” and what the x-axis “VL2-A” is.
Answer:
Figure 9a shows a histogram obtained directly from the Attune Cytometric Software. The x-axis is called VL2-A because it is the name of the channel used to collect the fluorescence values when excited with a violet laser at 405 nm. VL2-A is the green channel which detects fluorescence emissions in the range 507-537 nm. The y-axis, which is called counts, shows the number of cells that had the fluorescence value detected in VL2-A. We tried to clarify this now in the Materials and methods section.
- Fig. 9: Similar to Fig. 4, it is unclear whether there is difference among the various yeasts and the mutants. Please provide statistical tests for 9c and 9d to determine whether the membrane potential of different yeasts is different.
Answer:
In Figure 4 and Figure 9 we obtained data that indicate only a trend of change in fluorescence when adding the different effectors. Data were normalized with the ThT value (to 1.0) in order to obtain a fold value of fluorescence change. Because they are normalized data, which give us information about the tendency of changes, it is not possible to apply the suggested statistical analysis. This was done only with the data that helped us validate the changes due to CCCP and KCl for each strain.
The actual PMP numerical values can be seen in Tables 3 and 4, where through the accumulation of ThT a PMP numerical value could be obtained, and now, statistical analysis to obtain the information suggested by the reviewer is added.
- Please discuss why the result in Fig. 9d differs from that shown in Fig. 4e, even though both used ThT to measure membrane potential.
Answer:
In the Discussion it is mentioned that in the different techniques used throughout the work a similar trend in fluorescence changes was observed after the addition of the different effectors.
Particularly, in Figure 4e and Figure 9d a similar tendency in the changes but not in the fluorescence fold change magnitudes can be observed because the measurements in Figure 4e were taken after 40 min of incubation and with slight agitation (as indicated in Materials and methods section) and those estimated in Figure 9d were taken after 5 min of incubation.
Not to mention, Figure 4e was performed on a multiwell plate reader and Figure 9d on a flow cytometer.
The authors would like to thank Reviewer 1 for the valuable recommendations and for the time invested, which definitely enrich our text.
References for the response to Reviewer 1:
- Peña, A.; Sánchez, N.S.; Calahorra, M. The plasma membrane electric potential in yeast: probes, results, problems, and solutions: a new application of an old dye?. In Old Yeasts - New Questions.; Lucas, C., Pais, C., Eds.; IntechOpen: Rijeka, 2017, doi:10.5772/intechopen.70403
- Calahorra, M.; Sánchez, N.S.; Peña, A. Retraction Note to: Acridine yellow. a novel use to estimate and measure the plasma membrane potential in Saccharomyces cerevisiae. J. Bioenerg. Biomembr. 2019, 51, 313–313, doi:10.1007/s10863-019-09801-y.
- Padilla-Garfias, F.; Ríos-Cifuentes, L.; Sánchez, N.S.; Calahorra, M.; Peña, A. Study of the mechanism of ε-poly-L-lysine as an antifungal on Candida albicans and Saccharomyces cerevisiae. Biochim. Biophys. Acta - Gen. Subj. 2022, 1866, 130197, doi:10.1016/j.bbagen.2022.130197.
- García-Navarrete, M.; Avdovic, M.; Pérez-Garcia, S.; Ruiz Sanchis, D.; Wabnik, K. Macroscopic control of cell electrophysiology through ion channel expression. Elife. 2022, 11, e78075, doi:10.7554/eLife.78075.
- Peña, A.; Uribe, S.; Pardo, J.P.; Borbolla, M. The use of a cyanine dye in measuring membrane potential in yeast. Arch. Biochem. Biophys. 1984, 231, 217–225, doi:10.1016/0003-9861(84)90381-3.
- Peña, A.; Sánchez, N.S.; Calahorra, M. Estimations and actual measurements of the plasma membrane electric potential difference in yeast. In Membrane Potential: An Overview; Marušić, M., Ed.; Nova Science Publishers, Inc.: Hauppauge, New York, 2020; Vol. 1, pp. 81–104 ISBN 978-1-53616-743-6.
- Peña, A.; Sánchez, N.S.; Calahorra, M. Estimation of the electric plasma membrane potential difference in yeast with fluorescent dyes: comparative study of methods. J. Bioenerg. Biomembr. 2010, 42, 419–432, doi:10.1007/s10863-010-9311-x.
- Gitler, C.; Rubalcava, B.; Caswell, A. Fluorescence changes of ethidium bromide on binding to erythrocyte and mitochondrial membranes. Biochem. Biophys. 1969; 193(2):479-81. doi: 10.1016/0005-2736(69)90208-9).
- Peña et al., Multiple interactions of ethidium bromide with yeast cells. Arch. Biochem. Biophys. 1980, 201:420-428.
- Alexander A. Maskevich, Vitali I. Stsiapura, Valeriy A. Kuzmitsky, Irina M. Kuznetsova, Olga I. Povarova, Vladimir N. Uversky, and Konstantin K. Turoverov; Spectral Properties of Thioflavin T in Solvents with Different Dielectric Properties and in a Fibril-Incorporated Form; Journal of Proteome Research 2007 6 (4), 1392-1401
DOI: 10.1021/pr0605567

Reviewer 2 Report
The measurement of the membrane potential in microorganisms is an important research problem, since it allows one to characterize the state of the cell under different conditions. Therefore, this study is methodologically useful.
Some questions I would like to clarify:
- Fig.3. How would the curves look if KCl was added without CCCP ?
- Fig. 6. Figure 6 clearly shows the staining of mitochondria by thioflavine ( glucose ) in the cells of S. cerevisiae and C. albicans, while the cells of R. mucilaginosa and D. hansenii demonstrated no such staining. What could be the reason for this difference?
“The only exception to this behavior was R. mucilaginosa, which did not clearly show the fluorescence decrease upon the addition of KCl” - Is this related to the possible features of potassium transport in these yeasts?
Minor comment:
The names of yeast species need to be written in Italic
Author Response
REVIEWER 2
The measurement of the membrane potential in microorganisms is an important research problem, since it allows one to characterize the state of the cell under different conditions. Therefore, this study is methodologically useful.
Some questions I would like to clarify:
- Fig.3. How would the curves look if KCl was added without CCCP?
Answer:
When no CCCP is added (at the concentration we use), mitochondria are not involved in the response; the ThT that is inside the organelles, keeps there. Upon the addition of KCl, there is a slow increment in the fluorescence due to the direct activation of Pma1. Once the activation resumes, there is a gradual diminution of the fluorescence indicating the depolarization now exerted by the cation. The experiments (not shown in the manuscript) are displayed in the next figure:
(Please look at the attached file where there is a figure in the response)
In b) we can observe that if CCCP is added after KCl, a large increase in fluorescence arises as a result of the release of the dye from mitochondria.
Previously, it was reported that the Pma1 is stimulated 20 to 50% by the addition of 50 mM KCl [1,2], increasing the proton pump outside the cell and subsequently increasing the PMP, which in turn stimulates the transport of ThT that acts as a positive charge.
- Figure 6 clearly shows the staining of mitochondria by thioflavine ( glucose ) in the cells of S. cerevisiae and C. albicans, while the cells of R. mucilaginosa and D. hansenii demonstrated no such staining. What could be the reason for this difference?
Answer:
Mitochondria were not clearly observed with D. hansenii, M. guilliermondii and R. mucilaginosa, probably because their accumulation of the dye was to such extent that they cannot be clearly seen; however, the respective images of the same yeasts after adding CCCP showed a greater fluorescence (most clearly seen in the fluorometer traces), which is in agreement with the accumulation of the dye in these organelles.
Furthermore, in the discussion we mentioned that D. hansenii and R. mucilaginosa accumulate lipids, and must probably, ThT has some affinity for them; besides, R. mucilaginosa produces carotenoids that emit fluorescence in a similar band to that of ThT, so differentiating these organelles in this way is difficult.
- “The only exception to this behavior was R. mucilaginosa, which did not clearly show the fluorescence decrease upon the addition of KCl” - Is this related to the possible features of potassium transport in these yeasts?
Answer:
Sorry, but a decrease was observed upon the addition of KCl. The error has already been corrected in the text.
- The names of yeast species need to be written in Italic
Answer:
All the names of yeasts species and of genes were in italics in our original manuscript. Maybe, it was an error from the editorial when drafting into the template of the Journal. We are sorry for this, and it is now corrected and reviewed, thank you.
Finally, we would like to thank Reviewer 2 for the suggestions and observations that certainly are improving our manuscript.
References for the response to Reviewer 2:
- Andre Goffeau, Antoine Amory, Antonio Villalobo, and Jean-Pierre Dufour. The H+-ATPase of the yeast plasma membrane. Annals New York Academy of Sciences. 1982:91-98
- Anthony Ambesi, Manuel Miranda, Valery V. Petrov, Carolyn W. Slayman; Biogenesis and Function of the Yeast Plasma-Membrane H+-ATPase. J Exp Biol 1 January 2000; 203 (1): 155–160. doi: https://doi.org/10.1242/jeb.203.1.155

Round 2
Reviewer 1 Report
The authors have sufficiently addressed all the concerns raised in the last round of review. The reviewer would recommend the publication of this manuscript.